# Index of Body Inflammation for Maxillofacial Surgery Purpose-to Make the Soluble Urokinase-Type Plasminogen Activator Receptor Serum Level Independent on Patient Age



**Marcin Kozakiewicz [1],\*, Magdalena Trzcińska-Kubik [1] and Rafał Nikodem Wlazeł [2]**

[1] Department of Maxillofacial Surgery, Medical University of Lodz, 113 Żeromskiego Str., 90-549 Lodz, Poland; magdalena.trzcinska@gmail.com

[2] Department of Laboratory Diagnostics and Clinical Biochemistry, Medical University of Lodz, 251 Pomorska Str., 92-213 Lodz, Poland; rafal.wlazel@umed.lodz.pl

\* Correspondence: marcin.kozakiewicz@umed.lodz.pl; Tel.: +48-42-6393068

**Featured Application: Inflammation is still a threat to patients of all age groups. Monitoring the infection is essential to control it. The soluble urokinase-type plasminogen activator receptor (suPAR) is a very sensitive marker. Maxillofacial surgery is one of the fields of medicine widely fighting against infections. The concentration of suPAR holds an independent information for risk stratification of morbidity and mortality in various acute and chronic diseases. The use of an ultra-sensitive measure of the development of inflammation and at the same time independent on the patient's age would be clinically very beneficial.**

**Abstract:** Background: The serum suPAR level is affected in humans by it increases with age. Therefore it makes difficult interpretation and any comparison of age varied groups. The aim of this study is to find simple way to age independent presentation of suPAR serum level for maxillofacial surgery purpose. Methods: In generally healthy patients from 15 to 59 y.o. suPAR level was tested in serum before orthognathic or minor traumatologic procedures. Five ways to make the suPAR serum level assessment independent of age are proposed. Results: One way of making suPAR levels independent of age led to the result with the same statistical distribution as in the raw suPAR serum material and this distribution is the normal. Conclusion: The simple way for suPAR serum level analysis without its dependence on patient age is calculation of the index of body inflammation understood as square root of squared suPAR serum level divided by logarithm of patient age to base 10.

**Keywords:** age; inflammation; infection control; maxillofacial surgery; orthognathic surgery; traumatology; complications; medical diagnostics; soluble urokinase-type plasminogen activator receptor; suPAR

## 1. Introduction

The frequency of complications associated with aseptic and scheduled maxillofacial surgery is 1% [1] by 5% [2] up to reported 33% [3]. These data refer to young people and it is known that with age the risk of inflammatory complications increases. Inflammation is an important and one of the most common complications observed in this type of surgery.

When midface is operated on, the maxillary sinuses are filled with blood, which may be infected [4]. Postoperative swelling also needs to be considered after orthognathic surgery, as the result of the onset of an infection or a harmless postoperative reaction [5]. The only dangerous condition of mentioned series is a developing infection. It would be very valuable to know as early as possible about the inflammation. This would avoid disastrous symptomatic treatment with corticosteroids [6], which would suppress the symptoms of inflammation and allow the infection to develop secretly. In addition, it

would be worth knowing before starting the surgery whether a clinically healthy patient undergoing the scheduled procedure has no hidden inflammation.

The soluble urokinase-type plasminogen activator receptor (suPAR) is ultrasensitive marker aggressiveness of infectious conditions [7]. It was also proved that the concentration of suPAR holds an independent information for risk stratification of morbidity and mortality [8–12]. This 64 kDa protein, intensively studied since 1991 which Ploug et al. [13], found that agent derived from phorbol 12-myristate 13-acetate (PMA)-stimulated U937 cells having a high affinity for uPA, has one major disadvantage. This issue is reflected in clinical trials, making it impossible to compare different age groups of patients. Serum suPAR levels depend on the patient's age and, more worryingly, increase with age [14,15]. The sicker the patient becomes as age advances, the less clear the suPAR level is as an indication of inflammation going on in the patient's body. The level of this biomarker is high in sick patients but is also high in older patients.

While the acute-phase reactant C-reactive protein (CRP) is commonly used as the gold standard inflammation marker both in the clinic and in life-course research [16], suPAR is a newer biomarker of inflammation [17], which appears to be correlated with chronic rather than acute inflammation. Although CRP and suPAR are positively correlated, they appear to capture different aspects of inflammation [18]. CRP and IL-6 did not evidence consistent associations at age but suPAR increases [19]. Elevated suPAR was associated with accelerated pace of biological aging across multiple organ systems, older facial appearance, and with structural signs of older brain age [20]. The serum concentration of soluble uPAR correlates with inflammation and accelerated biological aging. Individuals with high levels of soluble uPAR have a greater decline in cognitive function and physical activity [21]. Moreover, suPAR is postulates as even be biomarkers of biological aging [14].

In the general population suPAR levels are higher in females and in young healthy individuals (a typical group of maxillofacial patients) are about 2.5 ng/mL (males) and 3.0 ng/mL (females). The suPAR level is affected by a lifestyle and risk factors and it increases with age [8,22,23]. Therefore it makes difficult any comparison of age varied groups. One would like to break free from this age-driven framework.

The aim of this study is to find simple way to age independent presentation of suPAR serum level.

## 2. Materials and Methods

192 patient were included into this study (114 females and 78 males). Age was $34.1 \pm 12.4$ year from 15 to 59 year old. All patients provided written informed consent and the study was approved by the University Ethic Committee (RNN/646/13/KB). Serum suPAR levels were measured in orthognathic and minor traumatological patients (i.e., inclusion criteria). Patients with cardiovascular problems, kidney disease and oncological issues were excluded (i.e., exclusion criteria).

The suPAR concentrations were determined with use of the suPARnostic ELISA, ViroGates (ViroGates, Banevænget 13, DK-3460 Birkerød, Denmark). Index of body inflammation was calculated in 5 modalities according to Equations (1)–(5):

$$\text{IBI}_{\text{suPAR/Age}} = 20\frac{\text{suPAR}}{\text{Age}} \tag{1}$$

$$\text{IBI}_{\text{suPAR/Age2}} = 500\frac{\text{suPAR}}{\text{Age}^2} \tag{2}$$

$$\text{IBI}_{\text{suPAR2/Age}} = 10\frac{\text{suPAR}^2}{\text{Age}} \tag{3}$$

$$\text{IBI}_{\text{suPAR2/lg\_Age}} = \frac{\text{suPAR}^2}{\log_{10}\text{Age}} \tag{4}$$

$$\text{IBI}_{\text{SQRT(suPAR2/lg\_Age)}} = 1.223 \sqrt[2]{\frac{\text{suPAR}^2}{\log_{10}\text{Age}}} \tag{5}$$

Data was analyzed in Statgraphics Centurion 18 (Statgraphics Technologies Inc. The Plains City, VA, USA). Simple regression of suPAR serum level and calculated indeces to patient age was performed. Kolmogorov-Smirnov test was done to compare distribution of detected suPAR in serum to index of body inflammation. Level of significance was established as $p < 0.05$.

### 3. Results

The results of suPAR levels and the distribution (Table 1) in investigated population was Gaussian one (normal distribution). It was confirmed that suPAR level depends on the patient age (cc = 0.31, $p < 0.0001$) (Figures 1 and 2).

The index constructed as suPAR/Age in relation to age has even stronger relation (cc = −0.75, $p < 0.0001$). The same moderately strong relation is for the index suPAR/Age^2 to age (cc = −0.80, $p < 0.0001$), and even the numerator of the fraction is more strong: suPAR^2/Age (cc = −0.50, $p < 0.0001$).

**Table 1.** Index of body inflammation calculated in five investigated ways. Summary view for distribution and age dependency.

| IBI Modality Name | Avarage ± SD | Stnd. Skewness [1] | Stnd. Kurtosis [2] | Distribution |
|---|---|---|---|---|
| suPAR (raw data) | 2.69 ± 0.57 ng/mL | 0.796 | 0.501 | Normal |
| suPAR/Age | 1.77 ± 0.72 | 5.428 | 0.930 | |
| suPAR/Age^2 | 1.66 ± 1.25 | 7.324 | 3.056 | |
| suPAR^2/Age | 2.45 ± 1.28 | 7.898 | 8.017 | |
| suPAR^2/lg_Age | 5.03 ± 2.03 [3] | 3.928 | 2.619 | |
| SQRT(suPAR^2/lg_Age) | 2.69 ± 0.56 [3] | 0.141 | 0.503 | Normal |

[1] Standardized skewness within range from −2 up to +2 means normal distribution. [2] Standardized kurtosis within range from −2 up to +2 means normal distribution. [3] Age independent variable ($p < 0.05$). IBI–index of body inflammation. SD–standard deviation.

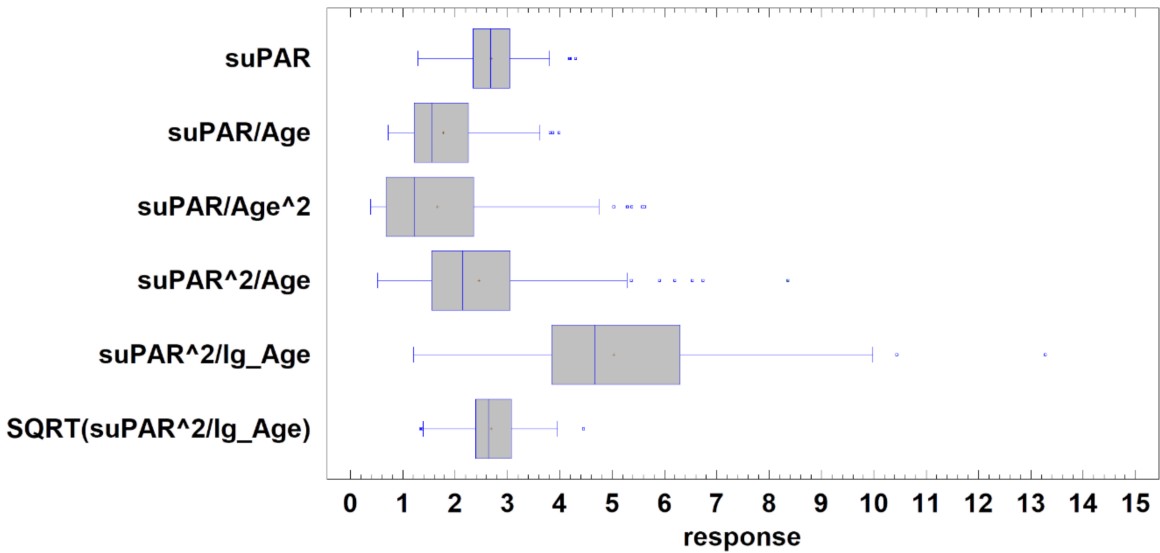

**Figure 1.** Soluble urokinase-type plasminogen activator receptor (suPAR) level in serum (ng/mL) and calculated body inflammation indeces (no units) separating the measure from patient age influence (average value is marked by red cross, median is marked by vertical line inside the boxes). Note: similarity of SQRT(suPAR^2/lg_Age) to raw data. All calculations performed according Equations (1)–(5).

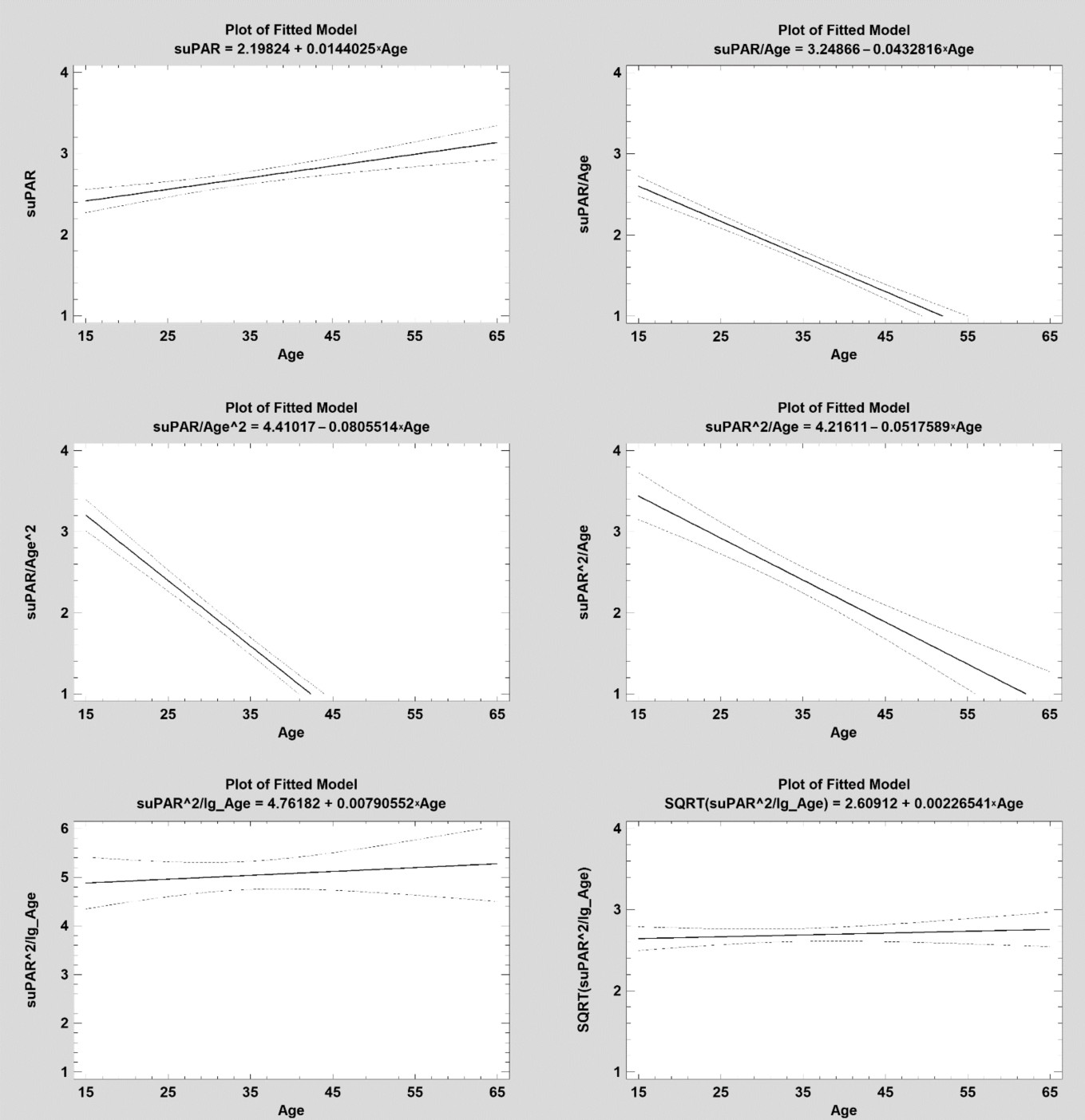

**Figure 2.** Relation of the soluble urokinase-type plasminogen activator receptor (suPAR, ng/mL) and calculated indeces to patient's age. Simple regression of suPAR raw data and patient age (top, left-hand side). Dash line describes confidence limits at level of 95%. The level of suPAR gradually and significant increases in time (years). The both lowest plots confirm age independence and only lower right-hand belongs normal distribution. Moreover, range of value SQRTsuPAR^2/lg_Age is similar to raw data range. Note: The calculations were based on indeces calculated according to the Equations (1)–(5).

Contrary to above (Figure 1), if the influence of the age (denominator) to the index is weaker, the index becomes independent on patient age (Figure 3): as suPAR^2/lg_Age (cc = 0.05, *p* = 0.5046) as well as SQRTsuPAR^2/lg_Age (cc = 0.05, *p* = 0.4861). The run of the regression plot is approximately horizontal.

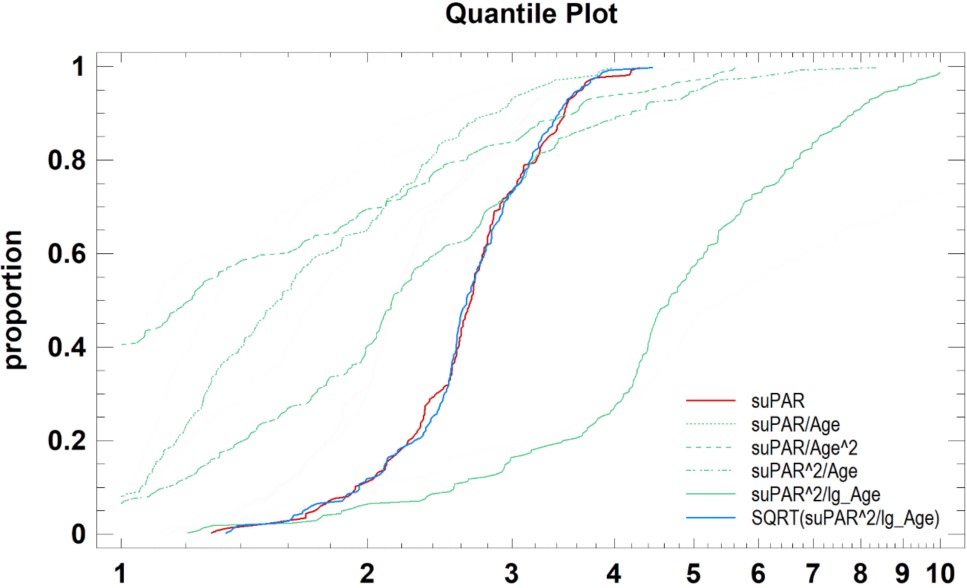

**Figure 3.** Values of calculated indeces. The search for body inflammation index-to make the suPAR level independent on patient age together with similar representation of variability. Accurate presentation of differences in suPAR distribution (age dependent variable: cc = 0.39; *p* < 0.0001) and a series of indeces distributions that make the result independent of the patient's age (as SQRTsuPAR^2/lg_Age calculated). Note: The calculations were based on indeces calculated according to the Equations (1)–(5).

There is not normal distribution noted in the first index of this twins. Finally, standardization of body inflammation index (IBI) to form Equation (6):

$$\text{IBI} = 1.223 \sqrt[2]{\frac{\text{suPAR}^2}{\log_{10} \text{Age}}} \tag{6}$$

makes possible to reach distribution similar to raw suPAR (Figure 4), as well the normal distribution of values of the index, together with simultaneous independence on patient age (bottom right-hand side in Figure 2).

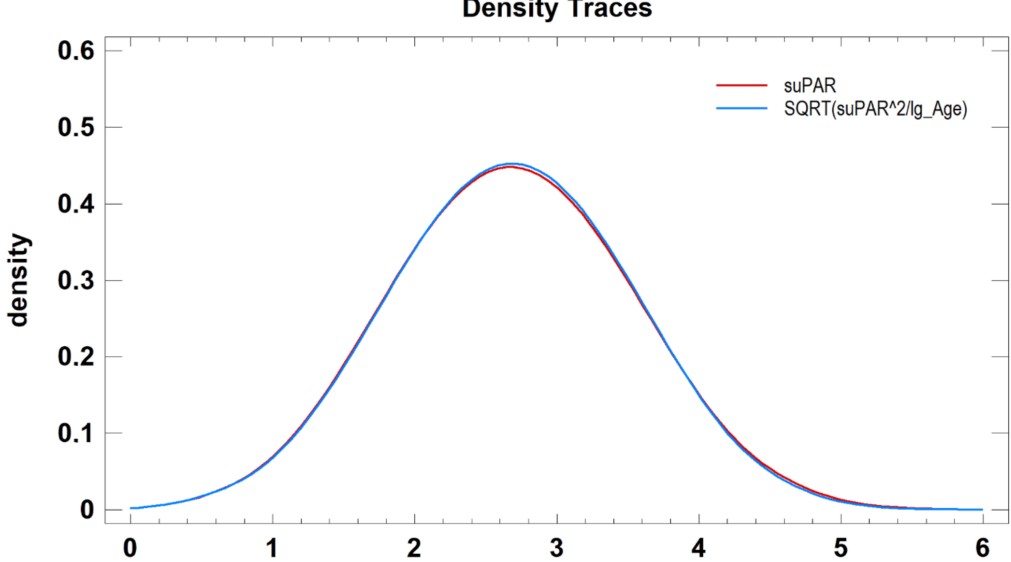

**Figure 4.** Raw suPAR distributions is the same as the Index of Body Inflammation (calculated as IBI$_{\text{SQRTsuPAR}^2/\text{lg\_Age}}$) distribution (*p* = 0.9110; Kolmogorov-Smirnov test 0.5613). In this way a measure of the inflammatory reaction (IBI) is obtained, independent on the patient's age, with a normal distribution and a range of values similar to the original serum activity test results. Note: The calculation was based on index derived from the Equation (6).

## 4. Discussion

Inflammatory postoperative direct complications after maxillofacial surgery have been already known [5,24]. It could be related to other indirect complications, such as rhinitis and empyema [2,25,26]. Postoperationally, the reaction in soft tissues is expected. However, its rapid and uncontrolled course may ruin the outcome of the treatment [27]. This is shocking for patients who associate aesthetic expectations with the result of the maxillofacial surgery. This can also be shocking for the maxillofacial surgeon when taking on extremely difficult challenges such as allogenic facial transplantation [28]. For this reason, authors believe that the greatest risk are patients with no inflammatory symptoms before the surgical treatment and already affected by a disease. Immune dysfunction or tiny inflammatory process are accompanied by a temporal shift in the innate and adaptive immune cells distribution, triggered by the overwhelming release of an inflammatory mediators from necrotic cells [27]. Anticipatory diagnosis would solve this issue. However, the diagnosis can be challenging as physical exam findings are absent initially, and become more noticeable much later in the disease course with decrease in mouth opening, wound dihescensing, soft tissue involvement, and pain with motion [28,29]. Searching, finding and proposing the use of new sensitive markers of the tiny or concealed pathological process gives hope to avoid these difficult clinical scenarios. One such marker is the soluble urokinase-type plasminogen activator receptor: suPAR.

The suPAR protein was found in 1991 as a marker of cancer progression. Up to these days several studies have shown the association of the protein with chronic diseases, including cardiovascular, renal, hepatic, and pulmonary diseases, and that it's level is an independent predictor of a negative outcome of various infectious diseases, sepsis, and in critically ill patients [30–33]. As the protein is expressed mainly on immune cells, suPAR reflects the level of chronic inflammation. Indeed the suPAR level reflects current inflammatory condition and therefore its role as predictive marker is known and studied across different fields of medicine. The protein itself seems not to have proinflammatory potential-it is released from membranes of activated (by acute-phase proteins) inflammatory cells. Across diseases, the suPAR level discriminates non-survivors from survivors [7,8]. In the general population an elevated suPAR level is associated with future development of cancer, cardiovascular diseases, and type 2 diabetes and is a predictor of premature mortality and renal failure [7,9,11]. There are no reports of suPAR levels in patients undergoing maxillofacial surgery in the available literature.

The barrier to the use of this emerging chronic inflammation marker in clinical conditions not life-threatening, but nevertheless threatening with other complications is its dependence on the patient's age. Thus, it is not only the existence of inflammation causes an increase in serum suPAR levels. The older the patient, the physiologically higher the suPAR level. suPAR levels increased from 2.39 ng/mL at age 38 to 3.01 at age 45 years [20]. Elevated suPAR is associated with accelerated pace of biological aging across multiple organ systems, older facial appearance, and with structural signs of older brain age. The observation of our series of patients confirms this report by interpolation (Figure 2: suPAR = 2.19824 + 0.0144025 × Age) in this study: 2.49 ng/mL for 20 y.o., 2.63 ng/mL for 30 y.o., 2.77 ng/mL for 40 y.o. and 2.92 ng/mL 50 y.o. patient. It is therefore impossible to give a universal suPAR level standard for healthy but age-different maxillofacial patients.

Initiating mathematical analyses of internal variability and trying to make the diagnostic measure independent of the patient's age, the formula for body mass index (BMI) was used. BMI is well standardized medical index. It is calculated as body weigh divided by squared patient height. Then it informs about body mass measure independently on the patient height. As the same method would be used for suPAR liberation from the patient age influence, an inconveniences appear. Simple division of the suPAR value by patient age (suPAR/Age) leads to reversal relationship, but still the index is related to the age. Body mass index (BMI) calculation method (suPAR/Ageˆ2) also supports the index relation to the patient age. Furthermore, the result has low value and normal distribution is lost (Table 1 and Figure 2). The description IBI as suPARˆ2/Age makes higher numerator of

fraction to change the values and distribution of index, but the index is still age dependent. Next, the logarithm in denominator of fraction makes the index independent on patient age (suPAR^2/lg_Age), and finally normalization by square rooting reduces IBI to range and distribution similar to raw suPAR register as serum suPAR level (Figure 4) giving patient age independence.

Due to the young population, the frequency of inflammatory complications after orthognathic surgery is low, which is even more surprising for the treatment team. Other issues include contact with the airways and digestive tract and the proximity of numerous sensory organs. The last issue is the invasiveness of the maxillofacial procedures in the facial skeleton: the mandible is divided into 3–4 segments (in sagittal osteotomy with genioplasty), the maxilla is separated from the base of the skull, the periosteum is elevated from large areas of the nose walls, not to mention the Le Fort III osteotomy [34]. The possibility of potentially dangerous complications in patients with a hidden medical problem (blindness [35,36], dyspnea, anosmia, apallic syndrome [37], dyphagia, dysgeusia) is high. Orthognathic maxillofacial procedures are a common surgery with set of complications [38,39]. Wound infections following orthognathic surgery occurs in 1.4 to 33.4% of patients, what are a major concern for surgical teams [40,41]. With the arms of the ultra-sensitive inflammatory marker independent of age, it increases the confidence to prepare patients for scheduled aseptic surgeries within the maxillofacial skeleton. When screening is performed, clinically mute patients with inflammatory, immunological or allergic dysfunctions will be separated and those patients will be treated causally before surgery. This will certainly protect patients from unexpected complications.

It should be noted that in this study the included patients treated as planned. Cochort includes mainly orthognathic patients with a small proportion of minor injuries. It seems worthwhile to check in the future how serum suPAR levels behave in maxillofacial patients, but after much higher trauma [42] and emergency surgical interventions.

A minor limitation of this study is the inclusion of maxillo-facial patients only. The results cannot be generalized to other areas of medicine at this stage. However, one problem has been solved for maxillofacial surgery. It is now possible to compare groups of patients differing in age and to detect older patients with an increased risk of inflammatory complications.

## 5. Conclusions

The simple way for suPAR serum level analysis without its dependence on patient age is calculation of the index of body inflammation (IBI) understood as square root of squared suPAR serum level in [ng/mL] divided by logarithm of patient age in years to base 10. The index calculated in this way can be used in maxillofacial surgery, especially in orthognathic patients.

**Author Contributions:** Conceptualization, M.K.; methodology, M.K.; software, M.K.; validation, M.K., and R.N.W.; formal analysis, M.K., and R.N.W.; investigation, M.K.; resources, M.K.; data curation, M.K.; writing—original draft preparation, M.K., and R.N.W.; writing—review and editing, M.K., and R.N.W.; visualization, M.K.; supervision, M.K., and R.N.W.; project administration, M.T.-K.; funding acquisition, M.K. All authors have read and agreed to the published version of the manuscript.

**Funding:** This research was funded by Medical University of Lodz grant number: 503-1-138-01-503-51-001-17, 503-1-138-01-503-51-001-18 and 503-1-138-01-503-51-001-19-00.

**Institutional Review Board Statement:** The study was conducted according to the guidelines of the Declaration of Helsinki, and approved by the Ethics Committee of Medical University of Lodz (RNN/646/13/KB, date of approval: 24.09.2013).

**Informed Consent Statement:** Informed consent was obtained from all subjects involved in the study.

**Data Availability Statement:** Not applicable.

**Conflicts of Interest:** The authors declare no conflict of interest.

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
