# Peer review of "Index of Body Inflammation for Maxillofacial Surgery Purpose-to Make the Soluble Urokinase-Type Plasminogen Activator Receptor Serum Level Independent on Patient Age"

_applsci, doi:10.3390/app11031345_

Round 1
Reviewer 1 Report
Dear Authors,
First, I found your research idea very interesting and I read your article two times, some parts even three times. Unfortunately, I did not understand what is “1.223” in the formula (5). It is completely unclear how you got this exact number (1.223). There are also some other inconsistencies.
Table 1. is missing results for formula (4).
There is no explanation of numbers “20”, “500” and “10” in formulas 1, 2 and 3.
Figure 4. has no “1.223” (still unclear what is “1.223”)
You need to improve methods as well as discussion.
Did you compare before and after surgery levels of suPAR?
Author Response
Please, find attached the rensponse for revision.
Author

Reviewer 2 Report
Dear editor,
research was written according to the instructions. The topic is contemporary and significant for practical clinical work. Materials and methods are described well.
Recommendation for correction:
1. Page 3 "The results of suPAR levels and the distribution (Tabale 1) in investigated population was Gaussian one (normal distribution)" - need to be corrected Table 1 instead Tabale 1
2. References should be written according to the instructions.Best regards
Author Response

(The authors gave the same response as above.)

Reviewer 3 Report
The study of Kozakiewicz M. and collaborators aims to find a simple way to predict maxillofacial surgery consequences focusing on suPAR serum level as biomarker.
- The abstract should be improved and better structured, in particular the aim of the study.
- Introduction should be expanded and better written. Information regarding the current tests available, ranging from hematological markers of infection and inflammation and eventually histology analysis should be cited.
It is not clear if the main complications associated with maxillofacial surgery are related to infection or inflammation or both. Please better clarify this part in the introduction.
- In material and methods authors assess: “Age was 34.1±12.4 year from 15 to 59 year old”. Please make this clear because it seems not correct.
- Authors should show other markers of inflammation, such as the canonical inflammatory markers IL-6, TNF-a and C-reactive protein and a correlation of these markers with plasmatic suPAR should be performed. Moreover to assess if this biomarker can be very suitable to the diagnosis of maxillofacial complications, it would be interesting to evaluate CCL2, a main chemokine involved in monocyte-macrophage action, and correlate this with suPAR.
- The discussion should be improved, the results obtained better discussed and compared to other similar findings in the literature. Same for the conclusion, please better write it.
- It is unclear if suPAR actually exerts a proinflammatory role or it just reflects an inflammatory condition. Please discuss this point.
- Personalized medicine is an important actual topic. Should these findings provide information regarding patients’ age on clinical decision-making?
- Please correct some typo in the text, for example in the results section “Tabale 1” correct into Table 1
Author Response
Please, find the rensponse for review attached.
Author

Round 2
Reviewer 1 Report
Dear Authors,
The amended article has cleared things up and statements compared to the first version.
Reviewer 3 Report
Thank you for the changes you made to the manuscript and for the comments provided. The manuscript is improved and now more of interest. However I would like to encourage the authors to revise the language as several sentences are not corrected.